# Analysis of Spatial—Temporal Variation in Ecosystem Service Value in Shandong Province over the Last Two Decades

Ting Li [1], Donghui Shi [2], Shuguang Jiang [2], Yu Li [2,*] and Huilu Yu [1,*]

1   School of Resources and Environmental Engineering, Ludong University, Yantai 264025, China; 17863607875@163.com
2   Institute of Geographical Sciences and Natural Resources Research, Chinese Academy of Sciences, Beijing 100101, China; shidonghui@igsnrr.ac.cn (D.S.); ginger_97@163.com (S.J.)
*   Correspondence: liy@igsnrr.ac.cn (Y.L.); yuhuilu73@163.com (H.Y.)

**Abstract:** During the rapid economic development process in Shandong Province, land use changes have led to the degradation of ecosystem service functions. Based on land cover data for the years 2000, 2010, and 2020, this study analyzes the spatiotemporal variation in land use and the corresponding ecosystem service value in Shandong Province. The research results reveal the following: (1) From 2000 to 2020, land use change is characterized by an increase in construction areas and water and a decrease in unused land, forestland, grassland, and cultivated land areas. Cultivated land was converted into construction land and grassland, and construction land was partially transformed into cultivated land in Shandong Province. The changes in land use types were more pronounced in the first decade compared to the later decade. (2) During the study period, the highest ESV among the primary service functions was observed in the regulating services, with hydrological regulation providing the highest ESV among the eleven secondary service categories. The spatial distribution of ESVs in Shandong Province is uneven. The high-value areas are concentrated in the Southern Four Lakes in Shandong Province and around Laizhou Bay. There was low ESV around mountainous areas in central Shandong Province. (3) Within the three time periods, cultivated land, grassland, and unused land provided a higher negative contribution rate, while water provided a higher positive contribution rate. Water had the highest positive contribution rate to ESV, while grassland and unused land had a higher negative contribution rate. Therefore, during the process of land development and utilization, it is important to pay attention to the impact of land use changes on the ecosystem, optimize the land use structure, restore ecologically fragile areas, and promote the sustainable development of the ecosystem and the economy.

**Keywords:** land use change; ecosystem service value; Shandong Province; spatial and temporal distribution; spatial variation



## 1. Introduction

Ecosystem services are key elements supporting the functioning of systems for Earth's life support [1]. ESV provides a material and nonmaterial basis for human survival and development. At the same time, ESV serves as a bridge between economic and social systems and ecosystems [2]. Land use is defined as the impact of human utilization patterns and conditions on the natural attributes of land, LUCC, due to the irrational human management of land. It is a major cause of global change [3,4] and a major driver of changes in the value of ecosystem services [5]. One can conduct spatial and temporal correlation studies on land-use dynamics and the value of ecosystem services, and identify trends in regional ecosystems. The above practices are not only conducive to the optimal adjustment of land use patterns, but also promote the sustainable development of socioeconomic–ecological systems.

As a result of rapid economic development, sustained population growth, and rising living standards in most countries around the globe; increasing human demand for ecosystem provisioning that triggers the decline of ecosystems; and a significant decline in the value of ecosystem services, many environmental problems have emerged over the last two decades, such as the pollution of water resources, the reduction of biodiversity, and land degradation [3,4,6]. Since ecosystem services are related to human well-being, such research has become popular, and most geographers and biologists are concerned [7].

An examination of prior research underscores the predominant themes within the realm of ecosystem service investigations. Primary among these is the evaluation of both the supply and demand of ecosystem services, along with an analysis of the equilibrium between the two. Notably, the techniques employed for estimating ecosystem service supply have achieved a considerable level of maturity. However, there remains a discernible scarcity in the literature concerning the comprehensive exploration of the equilibrium relationship between ecosystem service demand and supply [8–10]. The second is the characterization of the spatial and temporal evolution of ecosystem service values and scenario prediction, such as studies related to the coupling relationship between ESV and urbanization, ecological restoration zoning, and human well-being [11–13]. The third is research on the relationship between land use change and the value of ecosystem services. For example, Yang et al., Wei et al., Blumstein et al., and Zhang et al. explored the impacts of the quantitative and temporal characteristics of LUCCs on the dynamic evolution pattern of ESVs in Nanchang City; Dezhou City; MA, USA; and the Beijing–Tianjin–Hebei Rim, respectively [5,14–16]. Much of the above literature focuses on analyzing changes in the temporal and quantitative aspects of land use types. Fewer studies analyze the relationship from a spatial perspective. There is also less literature analyzing the contribution of LUCCs to ESV. Some scholars made attempts, such as Guo et al. and Shu et al., who used geospectrums of land use changes and hot spot analysis to quantitatively analyze the effects of quantitative and spatial–temporal changes in land use types on ESV [17,18]. The research on ESV and LUCCs started at the end of the last century and is mostly from the quantitative and temporal perspectives, and there is no effective methodology for research from the spatial perspective. Referring to the above literature, this paper tries to analyze the spatial–temporal characteristics of LUCCs with the dynamic degree model of land use, a diversity index, and the land use transfer matrix, and adopts hot spot analysis and assessment methods for the valuation of ecosystem service functions to reveal the spatial dynamic change characteristics of both.

In terms of study scales, the scales used to assess ESVs range from large and medium scales, such as national [19], provincial [20,21], and urban agglomerations [22], to small scales, such as watersheds [23,24] and typical regions [25–27]. The scales tend to be scale-dependent, and conducting ESV assessments at these scales has the disadvantage of under-representing the spatial information. The lack of attention to the microscale in the existing studies has led to crude results in ESV assessments. The scale is a key factor influencing ESV assessment. The results obtained using the grid scale are more refined than the administrative district scale and typical areas [28]. Taking complete provincial districts as the research unit, which have consistent institutional policies and similar socioeconomic activities within them, is conducive to analyzing the impact of institutional policies on LUCCs and ESV.

Shandong Province ranks highly in terms of its economy in the eastern coastal region of China. Since the reform and opening up of China, the rapid development of the economy and rapid increase in the level of urbanization have brought an enormous pressure to resources and the environment. Studying the spatial and temporal dynamics of the ecosystem service value in Shandong Province has helped to practice the "two mountains" theory and promote the construction of ecological civilizations in the last two decades. In this study, 1762 grids of 10 km × 10 km were created based on three periods of Landsat TM/ETM land use remote sensing image data for Shandong Province: in 2000, 2010, and 2020. Land use change dynamics and diversity indices were used to portray the land use types, quan-

tities, and diversity aspects in Shandong Province for the years 2000, 2010, and 2020. We referred to the table of equivalent value of ecological services per unit area by Xie et al. [29], corrected according to the local land use type and socioeconomic situation. Based on the consideration of the values of various types of ecosystems in Shandong Province, the dynamic changes and spatial–temporal differentiation of the value of ecosystem services at the grid scale in Shandong Province are explored in order to provide scientific reference for the rational utilization of land, protection and restoration of ecosystems, and provide scientific support for the promotion of ecological and economic sustainable development and the enhancement of people's well-being in Shandong Province.

## 2. Data Sources and Research Methodology

### 2.1. Overview of the Study Area

Bordering the Bohai Sea and the Yellow Sea, Shandong Province has a total coastline of 3505 km, with 16 prefectural-level cities such as Jinan and Qingdao and 136 county-level political districts, with a total land area of 158,800,000 square kilometers and a resident population of 101.53 million (Figure 1). The terrain of the territory is high in the center and low in the surroundings, with Mount Tai being the highest point in the province. The landforms are diverse, including the northwest Lu plains, Luzhong mountains, Ludong hills and other large geographical units. The climate type is a temperate continental monsoon climate, with rain and heat at the same time and sufficient sunshine. The unique geographic location and complex, varied landforms have given rise to diverse ecosystems and a complete range of land use types. These primarily include farmland, forests, wetlands, and marine ecosystems. In addition, Shandong is rich in agricultural products and has a full range of industrial sectors, making it a populous and economic province. Over the past 20 years, Shandong's urbanization level has increased rapidly, from 38.15% to 63.05% in 2020.

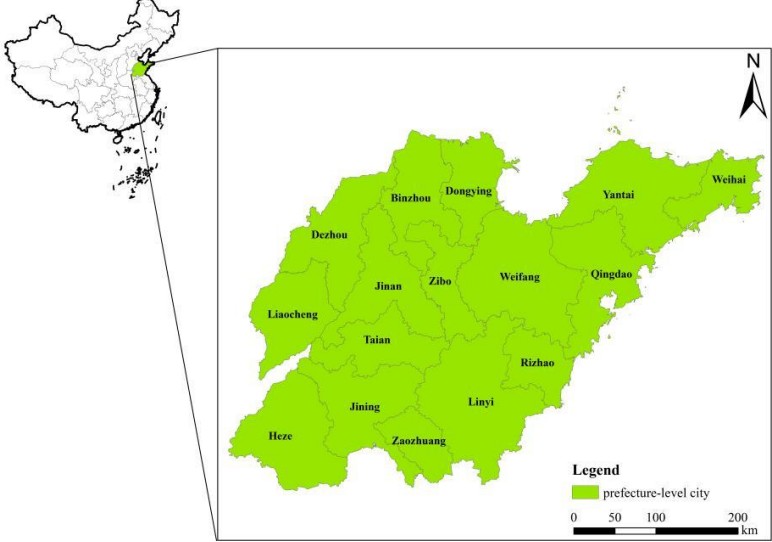

**Figure 1.** Geographical location map of the study area.

### 2.2. Data Sources

The Landsat TM/ETM land use remote sensing image data of Shandong Province in 2000, 2010, and 2020 were obtained from the Data Center for Resource and Environmental Sciences of the Chinese Academy of Sciences (http://www.resdc.cn, accessed on 23 May 2023), at a spatial resolution of 1 km × 1 km. Referring to the National Standard Classification of Land Use, this study adopts the first level of land types, which mainly includes six types of cultivated land, forestland, grassland, water, construction land, and unused land. The socioeconomic data used in this paper come from the 2000, 2010, and 2020 Statistical Yearbook of Shandong Province and the National Compendium

of Agricultural Product Cost and Benefit Information. Administrative district boundary data were obtained from the National Basic Geographic Information Service website (https://www.webmap.cn/main.do?method=index accessed on 23 May 2023).

*2.3. Research Methodology*

2.3.1. Attitudinal Modeling of Land Use Change Dynamics

The attitude of land use change momentum is a percentage reflecting the rate of land use change, which characterizes the degree of impact of human activities on the natural environment. In order to express the land use change, this paper uses the land use change dynamic attitude model and diversity index to calculate the amount of change in a land type during the study period [5,30], which is calculated using the following formula:

$$K = \frac{U_b - U_a}{U_a T} \times 100\% \tag{1}$$

$$Q_m = 1 - \sum_{i=1}^{n} x_i^2 / \left( \sum_{i-1}^{n} x_i \right)^2 \tag{2}$$

$U_a$, $U_b$ are the areas of any land use type in the study area in the base and end years, respectively; T is the time scale of the study; $K$ is the land use dynamic; $Q_m$ is the diversity index; and $x_i$ is the area of the first land use type in the study area.

2.3.2. Land Use Transfer Matrix

The land use transfer matrix can describe the transformation of regional land use types between the base and end periods of the study, and can reveal the direction and nature of the transfer between land use types [31], which is calculated as follows:

$$S_{ij} = \begin{bmatrix} S_{11} & S_{12} & S_{13} & \dots & S_{1n} \\ S_{21} & S_{22} & S_{23} & \dots & S_{2n} \\ S_{31} & S_{32} & S_{33} & \dots & S_{3n} \\ \vdots & \vdots & \vdots & \vdots & \vdots \\ S_{n1} & S_{n2} & S_{n3} & \dots & S_{nn} \end{bmatrix} \tag{3}$$

where $S$ is the area of each category, $n$ is the number of land use types, and $i$, $j$ are the land use types at the base and end of the study period, respectively.

2.3.3. Methodologies for Valuing Ecosystem Services

Using Xie et al.´s [29] value equivalent scale of ecosystem services per unit area, the value equivalent scale was adjusted according to the land use types in Shandong Province. The 1/7 ecosystem coefficient correction method proposed by Xie et al. [32] was used and calculated as follows: Taking the three-year average yields of wheat, rice, and maize grain crops, the sown area, and the national average purchase prices of the three grains in 2015 (2.76 CNY·kg$^{-1}$, 2.33 CNY·kg$^{-1}$, and 1.88 CNY·kg$^{-1}$, respectively) in Shandong Province as the base data, the economic value of the function of the ecosystem of the farmland per unit area in providing food production services was calculated using Equation (4) to be 2001.21 CNY·hm$^{-2}$, and multiplying the obtained result by the ecosystem service value equivalent per unit area table, the ecosystem supply coefficient per unit area of ecosystems in Shandong Province could be obtained (Table 1). The construction land factor is approximated to be zero, so it is not considered here. The calculation formula is as follows:

$$E_a = \frac{1}{7} \sum_{i=1}^{n} \frac{m_i p_i q_i}{M} \tag{4}$$

where $E_a$ is the economic value of the function of providing food production services per unit area of farmland ecosystem (CNY/hm$^2$); $i$ is the type of farmland crop; $m_i$ is the $i$ area planted with the seed crop (hm$^2$); $p_i$ is the $i$ national average price of the seed crop in a

given year (CNY/t); $q_i$ is the $i$ yield per unit area of the seed crop (t/hm$^2$); and $M$ is the area planted with all crops (hm$^2$).

**Table 1.** Eco-services provision coefficient per unit area of ecosystem in the Shandong Province (CNY·hm$^{-2}$·a$^{-1}$).

| Primary Ecosystem Services | Secondary Ecosystem Services | Cultivated Land | Forestland | Grassland | Water | Unused Land |
|---|---|---|---|---|---|---|
| Provisioning services | Food supply | 1701.03 | 480.29 | 466.95 | 1310.79 | 10.01 |
| | Raw material supply | 800.49 | 1090.66 | 687.08 | 730.44 | 30.02 |
| | Water supply | 40.02 | 560.34 | 380.23 | 10,886.60 | 20.01 |
| Regulating services | Gas regulation | 1340.81 | 3582.17 | 2414.80 | 2671.62 | 130.08 |
| | Climate regulation | 720.44 | 10,736.51 | 6383.87 | 5893.57 | 100.06 |
| | Environment purification | 200.12 | 3211.95 | 2107.94 | 9155.55 | 410.25 |
| | Hydrological regulation | 540.33 | 8094.91 | 4676.17 | 126,546.74 | 240.15 |
| Supporting services | Soil conservation | 2061.25 | 4372.65 | 2941.78 | 3241.97 | 150.09 |
| | Maintaining nutrient cycling | 240.15 | 330.20 | 226.80 | 250.15 | 10.01 |
| | Biodiversity | 260.16 | 3982.41 | 2674.96 | 10,426.32 | 140.08 |
| Cultural services | Aesthetic landscape | 120.07 | 1751.06 | 1180.72 | 6624.02 | 60.04 |
| | Total | 8382.26 | 8024.87 | 38,193.16 | 24,141.31 | 177,737.78 |

The total supply of ecosystem services was calculated according to Equation (5).

$$ESV_f = \sum \left( A_k \times VC_{kf} \right) \tag{5}$$

$$ESV = \sum \left( A_k \times VC_k \right) \tag{6}$$

In Equations (5) and (6), $ESV_f$ is the supply of ecological service $f$; $VC_{kf}$ is the supply coefficient of ecological service $f$ for the $k$th land use type; $ESV$ is the total supply of ecosystem services in the study area; $A_k$ is the area of the $k$th land use type in the study area; and $VC_k$ is the ecosystem service supply of the $k$th land use type.

2.3.4. Hot Spot Analysis

The cold spot and hot spot analyses of ecosystem service value can be used to explore whether spatial variation in ESV is characterized by the phenomenon of clustering of high values (hot spots) and clustering of low values (cold spots), as well as to determine where clustering occurs spatially [33]. The formula is as follows:

$$
\begin{aligned}
G_i^* &= \frac{\sum_{j=1}^n W_{ij} X_j - X \sum_{i=1}^n W_{ij}}{s \sqrt{\left[ n \sum_{j=1}^n W_{ij}^2 - \left( \sum_{j=1}^n W_{ij} \right)^2 \right] / (n-1)}} \\
X &= \frac{1}{n} \sum_{i=1}^n X_i \\
S &= \sqrt{\frac{1}{n} \sum_{i=1}^n X_i^2 - (X)^{-2}}
\end{aligned}
\tag{7}
$$

Here, $n$ is the number of grids in the study area; $X_i$ is the ecosystem service value of grids $i$ and $j$, respectively; $X$ is the mean value; and $W_{ij}$ is the spatial weight matrix. If the amount of ESV change in a certain range is higher compared to the surrounding area, it is a hot spot area, which indicates that the ESV has increased more in that area; if the opposite is true, it is a cold spot area, which indicates that the ESV has decreased significantly in that area.

2.3.5. Contribution of Ecosystem Service Value

In order to explore the influence of changes in land use type on the value of ecosystem services, referring to the method of Zhang et al. [33], the contribution rate of the value of ecosystem services was used to reveal the main contributing factors and sensitive factors affecting the amount of change in ESV [34]. The formula is as follows:

$$S_i = \frac{\Delta ESV_i}{\sum_{i=1}^{n} |\Delta ESV_i|} \times 100\% \tag{8}$$

$S_i$ is the contribution of the $i$th land use type to the value of ecosystem services over the study period (%); $\Delta ESV_i$ is the change in the value of ecosystem services of the ith land use type over the study period ($10^8$ CNY); and $n$ is the number of land use types (n).

## 3. Analysis of Results
### *3.1. Land Use Structure Change*
3.1.1. Changes in Land Use Structure over Time

From 2000 to 2020, land use types in Shandong Province mainly consisted of arable land, forest land, watersheds, and construction land, with the sum of the areas of the four land types in the three periods accounting for 86.81%, 82.90%, and 81.75% of the total area of Shandong Province (Table 2). The area of construction land has increased by 931,676 $hm^2$ in the last two decades, accounting for 6.00% of the total area, making it the land use type with the largest increase. The area of water has increased by 993,168 $hm^2$ in the last two decades, accounting for 0.59% of the total area, which confirms that to a certain extent, the economic development of Shandong Province has not been at the expense of the ecological environment. In the past two decades, the area of arable land has decreased by 282,294 $hm^2$, which indicates that the protection of arable land in Shandong Province still needs to be further strengthened; forest land, grassland, and unused land have decreased by 554,784 $hm^2$, 537,371 $hm^2$, and 1,145,169 $hm^2$, respectively, with a cumulative decrease ratio of 4.75%, and the nonagricultural land has continued to decrease, which indicates that the degree of land use changes affected by human activities has been gradual. This indicates that land use changes are gradually influenced by human activities, and the pressure of ecological protection is continuously increasing; this is related to the occupation of other land types in the process of rapid urbanization and industrialization.

**Table 2.** Dynamic changes of land use in Shandong Province from 2000 to 2020.

| Year | Cultivated Land | | Forestland | | Grassland | | Water | | Construction Land | | Unused Land | |
|---|---|---|---|---|---|---|---|---|---|---|---|---|
| | Area /$hm^2$ | Proportions/% | Area /$hm^2$ | Proportions/% | Area /$hm^2$ | Proportions/% | Area /$hm^2$ | Proportions/% | Area /$hm^2$ | Proportions/% | Area /$hm^2$ | Proportions/% |
| 2000–2010 | −106,078 | −0.70 | −58,622 | −0.38 | −533,882 | −3.44 | 94,912 | 0.61 | 743,247 | 4.79 | −136,436 | −0.88 |
| 2010–2020 | −176,216 | −1.14 | 3838 | 0.03 | −3489 | −0.02 | −1745 | −0.02 | 188,429 | 1.21 | −8724 | −0.06 |
| 2000–2020 | −282,294 | −1.84 | −54,784 | −0.35 | −537,371 | −3.46 | 93,168 | 0.59 | 931,676 | 6.00 | −145,160 | −0.94 |

As shown in Figure 2, the degree of land use change during the period 2000–2010 is more drastic than that for 2010–2020. Construction land on the Jinan–Zibo–Weifang line increased significantly, crowding out a large amount of arable land; the expansion of construction land resulting from population pressure in the Linzhi metropolitan area is obvious, and part of the grassland transformed into arable land; in Jiaozhou Bay, there has been a rapid urbanization process, with obvious construction on land previously occupied by arable land; and in Laizhou Bay and Bohai Bay, there has been an increase in the size of the watershed area, and the area of unused land has reduced.

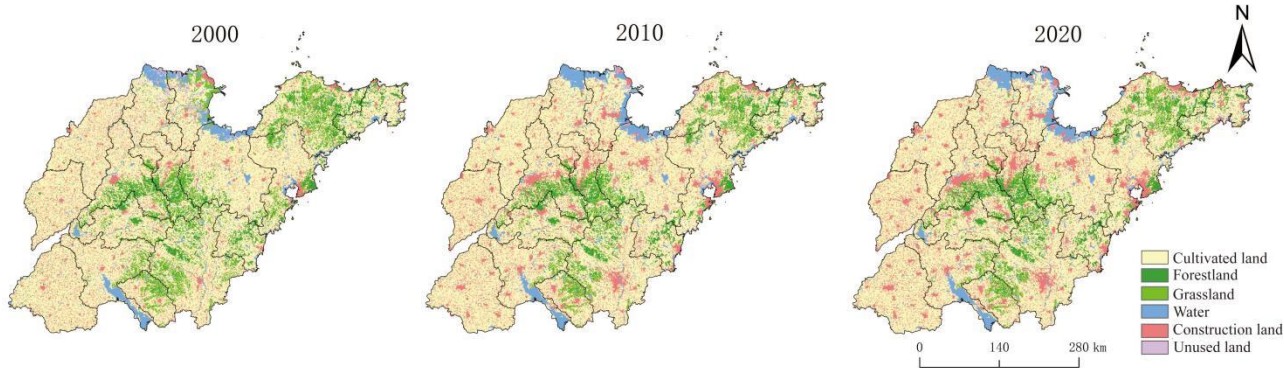

**Figure 2.** The map of land use in Shandong Province from 2000 to 2020.

3.1.2. Analysis of Land Use Change Dynamics and Diversity Indices

As can be seen in Figure 3, firstly, the dynamic fluctuation in the change in each land use type in Shandong Province during the period of 2000–2020 shows a polarized phenomenon, in which the magnitude of the changes in the areas of unused land, construction land, and grassland is large, which is more than 3.5%; and the magnitude of the change in the areas of cultivated land, forestland, and water is small, which is less than 2%. Secondly, the change in the area of each category from 2000 to 2010 is consistent with the situation from 2000 to 2020, and the overall land change from 2010 to 2020 is small. Specifically, from 2000 to 2010, unused land had the greatest motivation for change and water had the least. In the period 2010–2020, the greatest motivation for change was found in unused land and the least in forestland.

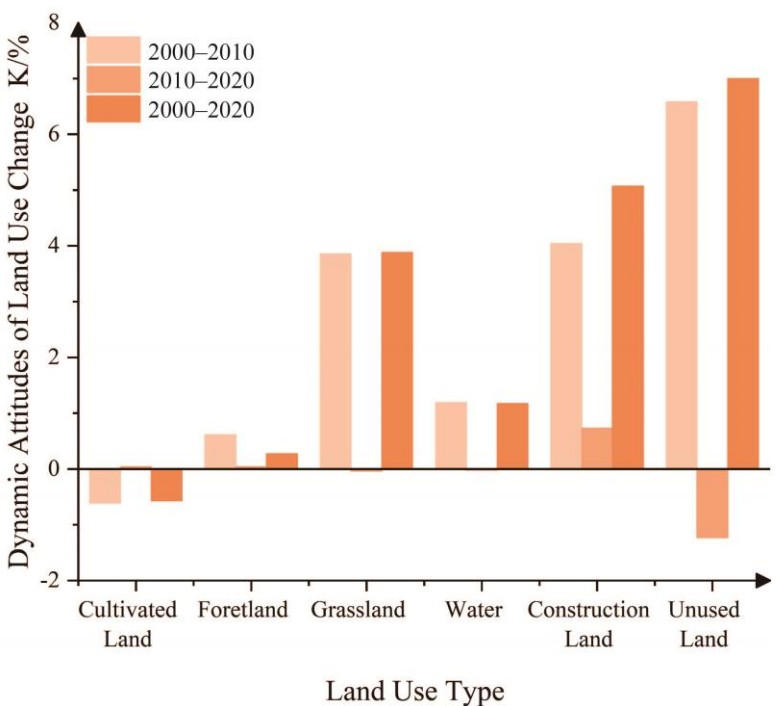

**Figure 3.** Dynamic attitudes of land use change in Shandong Province from 2000 to 2020.

The land use diversity index of Shandong Province in 2000, 2010, and 2020 was 0.5348, 0.5216, and 0.5253, respectively. The land use diversity indices of the three periods are all greater than 0.5, indicating that the land use structure of Shandong Province has been relatively stable over the past 20 years, and the degree of land diversification is still at a medium level, but the land use diversity indices have declined to a certain extent, of which

the most important reasons for this are the continuous increase in the area of construction land and the increase in the areas of forestland, grassland, and unused land occupied for construction land. This indicates that in the next land use process, Shandong Province should pay attention to the balance of land use and compensation, improve the land use structure, and increase the efficiency of land use.

### 3.1.3. Analysis of the Dynamic Land Use Transfer Matrix

As shown in Figure 4, from 2000 to 2010, the largest area transferred was construction land, mainly from cultivated land ($975.5 \times 10^3$ hm$^2$) and water ($76.5 \times 10^3$ hm$^2$), followed by cultivated land, which was mainly transferred from construction land ($404.7 \times 10^3$ hm$^2$) and grassland ($397.8 \times 10^3$ hm$^2$). The largest areas transferred out were cultivated land ($1115.4 \times 10^3$ hm$^2$) and grassland ($569.3 \times 10^3$ hm$^2$), which were mainly converted to construction land ($975.7 \times 10^3$ hm$^2$) and water ($96.9 \times 10^3$ hm$^2$). However, the area of construction land transferred to cultivated land ($404.7 \times 10^3$ hm$^2$) was much smaller than the area of cultivated land transferred to construction land ($975.7 \times 10^3$ hm$^2$). This indicates that in the rapid urbanization stage, human activities change the way that arable land and grassland are used, as well as advance the process of urbanization and industrialization to obtain economic benefits to meet human well-being needs.

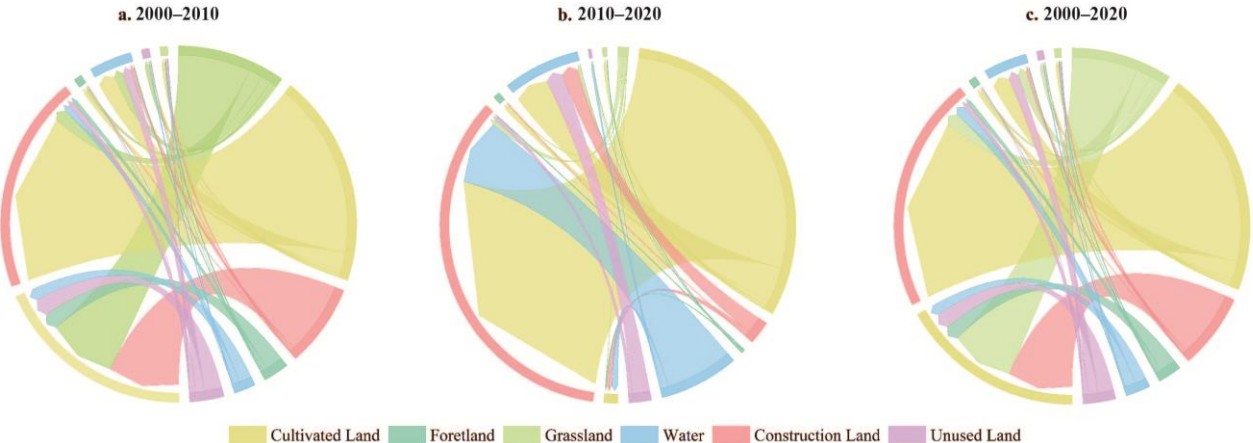

**Figure 4.** Land use change transfer matrix of Shandong Province in different periods.

In 2010–2020, the amount of land transferred is smaller in each category, with $191.8 \times 103$ hm$^2$ of arable land transferred to other land types, but the area is about 1/5 of the area transferred out in the previous period. The largest land type that saw changes was built-up land, which was developed from cultivated land ($166.1 \times 10^3$ hm$^2$) and water ($39.3 \times 10^3$ hm$^2$), followed by water ($42.9 \times 10^3$ hm$^2$), which was mainly transferred from cultivated land ($20.7 \times 10^3$ hm$^2$) and unused land ($10.3 \times 10^3$ hm$^2$). The amount of land transfers depends on the stringency of the land administration policy, i.e., the stricter the policy, the smaller the land transfers will be accordingly.

Throughout the entire 20-year period, the top three land categories in terms of area transferred were construction land ($1328.1 \times 10^3$ hm$^2$), cultivated land ($1038.3 \times 10^3$ hm$^2$), and water ($245.7 \times 10^3$ hm$^2$), which were mainly transferred from cultivated land ($1125.2 \times 10^3$ hm$^2$), construction land ($396.3 \times 10^3$ hm$^2$), and grassland ($391.6 \times 10^3$ hm$^2$) and water ($106.7 \times 10^3$ hm$^2$), respectively. The areas transferred out, in descending order, were cultivated land ($1276.4 \times 10^3$ hm$^2$), grassland ($571.2 \times 10^3$ hm$^2$), and construction land ($425.8 \times 10^3$ hm$^2$), and they were mainly converted to construction land ($1125.2 \times 10^3$ hm$^2$), cultivated land ($391.6 \times 10^3$ hm$^2$), and cultivated land ($396.3 \times 10^3$ hm$^2$). The transfer of arable land to construction land is the main cause of the loss of arable land. Compensation for arable land is provided by converting part of the grassland and building land to arable land. On the whole, cultivated land and construction land represent a large proportion of the transfer process on both sides, reflecting the fact that the conversion of the two types of

land is closely related to people's production and lifestyles. Rapid urbanization expands more urban construction land and crowds out space for cultivated land. At the same time, under the guidance of policies such as the "red line of arable land protection" and "green and coordinated development", part of the construction land, grassland, and other land categories have been supplemented with arable land, resulting in a certain increase in the area of cultivated land. However, it is still lower than the level in 2000, indicating that Shandong Province has a long way to go in protecting ecological land such as cultivated land and grassland.

### 3.2. Characterization of Temporal Changes in ESV

In terms of total ESV, the value of ecosystem services increased by $7.13 \times 10^8$ CNY in 2010 compared to 2000 and decreased by $16.73 \times 10^8$ CNY in 2020 (Table 3). In terms of the function of each ecological service, the value of the hydrologic regulation service was more stable in the three years and ranked first among the eleven secondary services; thereafter, it was the values provided by soil conservation, climate regulation, gas regulation, and biodiversity, respectively, that were the highest. The value of hydrologic regulation and water supply increased from 2000 to 2010, while the value provided by the remaining nine service functions decreased. The amount of change in the value of the supply of 11 categories of secondary services from 2010 to 2020 is negative, indicating an overall downward trend in the value of ecosystem services during this period.

**Table 3.** Values of and changes in ecosystem service in Shandong Province from 2000 to 2020. ($10^8$ CNY, %).

| Ecosystem Services | | 2000 | | 2010 | | | 2020 | | |
|---|---|---|---|---|---|---|---|---|---|
| | | ESV | Percentage | ESV | Percentage | Change 2000–2010 ESV | ESV | Percentage | Change 2000–2010 ESV |
| Provisioning services | FS | 197.17 | 6.69 | 193.82 | 6.56 | −3.35 | 190.80 | 6.50 | −3.02 |
| | RMS | 108.54 | 3.68 | 10.03 | 3.52 | −4.50 | 102.63 | 3.49 | −1.41 |
| | WS | 101.41 | 3.44 | 109.32 | 3.7 | 7.90 | 109.06 | 3.71 | −0.25 |
| Regulating services | GR | 227.90 | 7.73 | 213.84 | 7.24 | −14.06 | 211.47 | 7.20 | −2.37 |
| | CR | 313.27 | 10.6 | 277.58 | 9.4 | −35.68 | 276.39 | 9.41 | −1.19 |
| | EP | 154.45 | 5.24 | 149.23 | 5.05 | −5.22 | 148.73 | 5.06 | −0.50 |
| | HR | 1205.35 | 40.9 | 1294.85 | 43.8 | 89.50 | 1291.81 | 43.98 | −3.03 |
| Supporting services | SC | 321.77 | 10.9 | 304.19 | 10.3 | −17.58 | 300.55 | 10.23 | −3.64 |
| | MN | 33.13 | 1.12 | 31.69 | 1.07 | −1.44 | 31.27 | 1.06 | −0.42 |
| | CB | 185.47 | 6.29 | 178.28 | 6.04 | −7.19 | 177.69 | 6.05 | −0.59 |
| Cultural services | AL | 98.42 | 3.34 | 97.16 | 3.29 | −1.25 | 96.86 | 3.30 | −0.31 |
| Total | | 2946.86 | 100 | 2953.99 | 100 | 14.26 | 2937.26 | 100.00 | −33.46 |

Among the five major land categories, water contributes the highest value, constituting approximately 50% of the total value provided by all land categories (Figure 5). This suggests that water has a huge role to play in the ecosystem and generates a high economic value. This is followed by cultivated land, which provides about a quarter of the total value provided by all land types, followed by forestland and grassland, with unused land providing the lowest ESV. In terms of the year-to-year trend of ESV provided by each category, cultivated land, grassland, and unused land declined year by year; forest land first declined and then rebounded, but there is still a gap of $20.92 \times 10^8$ CNY; and water first rose sharply and then declined slightly.

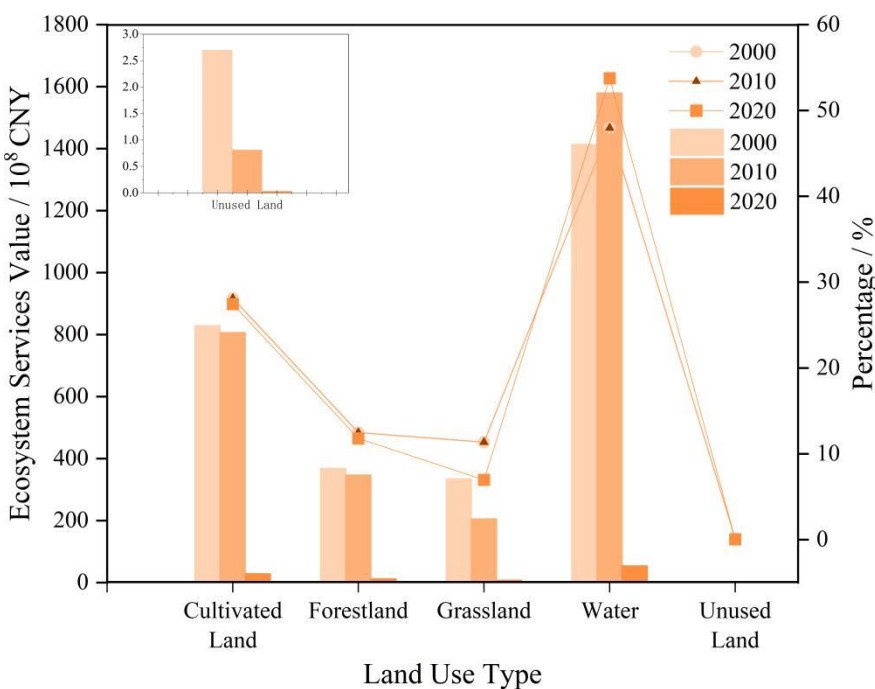

**Figure 5.** Values and changes of ecosystem service in Shandong Province from 2000 to 2020.

### *3.3. Patterns of Spatial Distribution of ESV and Characteristics of Change*

At the grid scale, the natural discontinuity point method was used to categorize ESV in the order from high to low as a high-value area, higher-value area, middle-value area, lower-value area, and low-value area, which resulted in the spatial distribution pattern of ecosystem service value in Shandong Province (Figure 6). In terms of the spatial distribution pattern, the high-value areas are concentrated in the northern areas of Binzhou City and Dongying City, the Laizhou Bay area, Huishan County in Jining City, and the western part of Dongping County in Tai'an City; the higher value areas are scattered in other areas except Dezhou City, Liaocheng City, and Zaozhuang City. The medium-value areas are distributed in the mountainous areas of central Shandong and the Shandong Peninsula. The lower-value zones are more dispersed and cross-distributed with the medium-value and low-value zones; the low-value zones are mainly distributed around the mountainous areas of central Shandong, and the ESV in its eastern part has been more distributed since 2010. Specifically, during the twenty-year period, the lower-value zones in the Shandong Peninsula region replaced the medium-value zones as the type with the largest distribution, and the number of grids in the lower-value zones in the mountainous areas of central Shandong gradually increased, which is related to the accelerated urbanization process and the increase in the area of land for construction. The ESVs of Shandong Province are mostly concentrated in the low-, lower-, and middle-value zones, indicating that the overall ESV of Shandong Province is low, and future development should focus on coordinating the relationship between economic development and ecological protection to improve the ESV. The raster calculator was used to calculate the amount of change in the three stages. The results are shown in Figure 7. The difference in the amount of change in the ESV between 2000 and 2010 is large, and the ESV in the northern part of the Yellow River Delta high-efficiency eco-economic zone in 2010 is greater than that in 2000, in addition to the fact that most of the areas have experienced a small decrease in their ESV after ten years of development. The amount of change in the latter stage is small, and the change from 2000 to 2020 is similar to that from 2000 to 2010. As shown in Figure 8, most of the hot spot areas from 2000 to 2020 are concentrated in the coastal areas of Binzhou, Dongying, and Weifang, as well as in the Nansihu area of Jining, and the hot spot areas in the three time periods first increase and then decrease, which is consistent with the changes in the ESV. The sharp

decrease in the cold spot areas indicates that the land use structure in west Lu has been optimized to a certain extent.

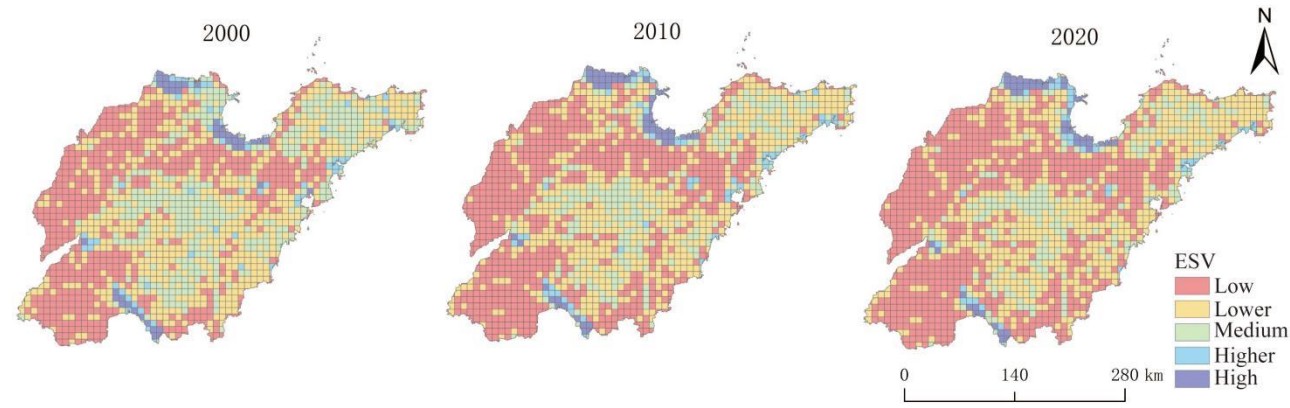

**Figure 6.** Spatial distribution of ESV in Shandong Province from 2000 to 2020.

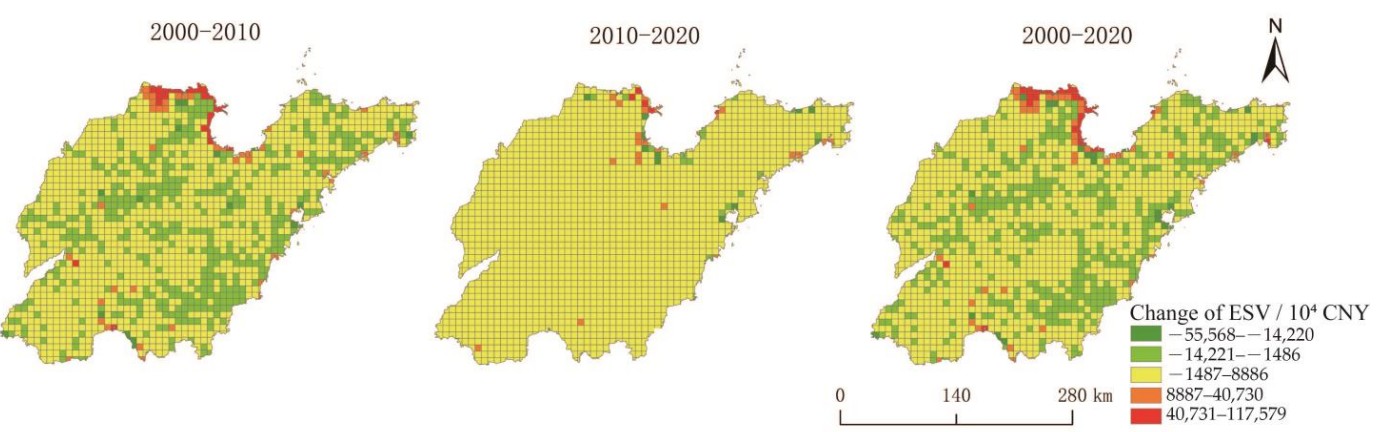

**Figure 7.** Amount of change of ESV in Shandong Province from 2000 to 2020.

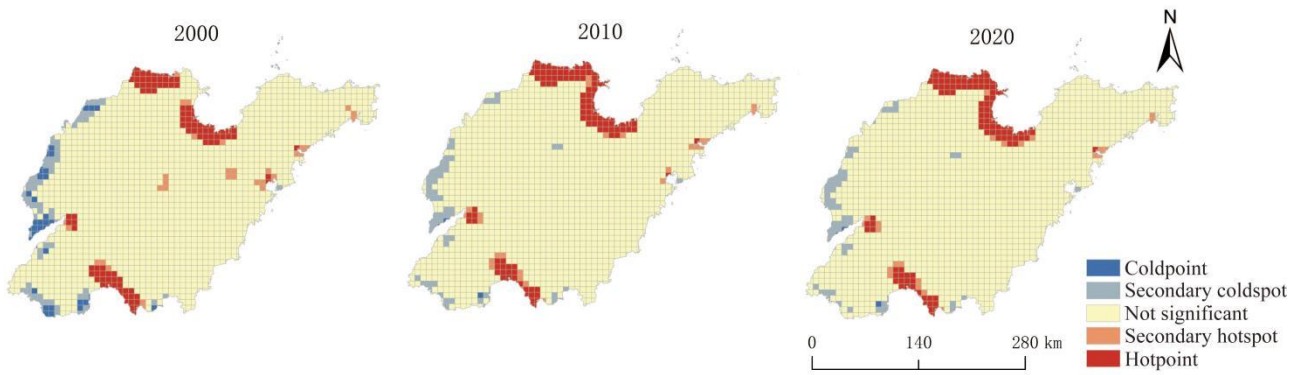

**Figure 8.** Spatial distribution pattern of cold and hot spots of ESV in Shandong Province from 2000 to 2020.

### 3.4. Contribution of Ecosystem Service Value

As shown in Figure 9, the change in the watershed area between 2000 and 2010 had the largest positive contribution to the ESV at 50.26%, and the largest negative contribution was from grassland (−38.40%). During the period 2010–2020, the change in the area of unused land contributed 71.91% to the negative ESV, which is the largest among all land categories, followed by cultivated land, which contributed 36.43% less than in the previous period. Overall, the changes in the areas of water (47.50%) and grassland (−37.00%) over

the twenty-year period contributed the most positively and negatively to ESV, respectively. Table 2 shows that grasslands have decreased in area significantly over the last 20 years and have a low contribution to the value of ecosystem services. This shows that while focusing on economic development, local governments should also pay attention to ecological land such as grassland and water, gradually restore the areas of grasslands, and scientifically and rationally plan the green development and utilization of waters so as to enhance their contribution to the ESV.

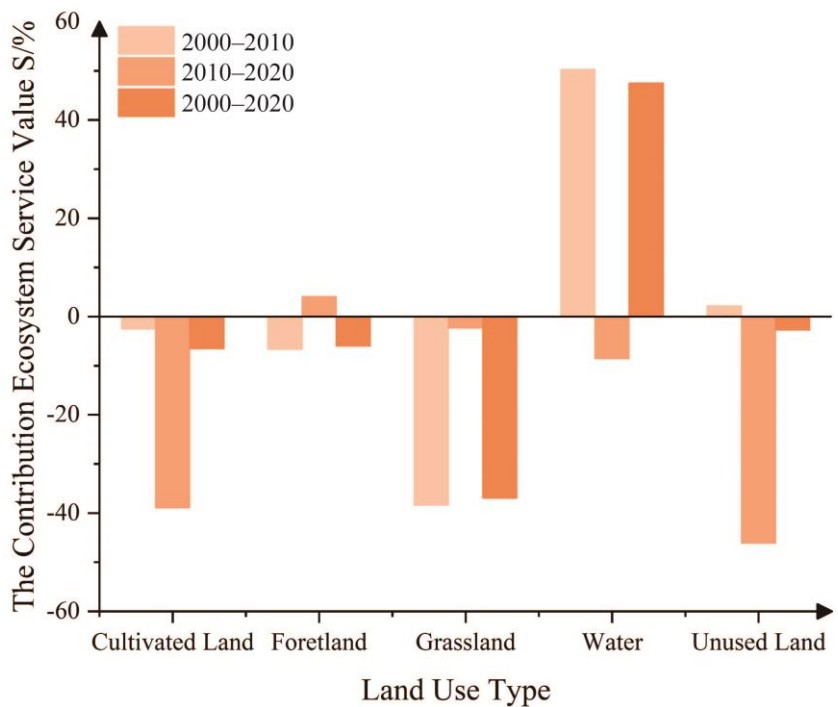

**Figure 9.** The contributions to ecosystem service value in Shandong Province from 2000 to 2020.

## 4. Discussion

### 4.1. Discussion

In this paper, the land use dynamic change degree, land use diversity index, and transfer matrix are used to analyze land use changes. The study concludes that land use change shows a clustered growth pattern. In the northern region of the central mountainous area in Shandong Province, Rizhao urban area, and Qingdao, the growth trend of urban construction land area is obvious, and the center of the county in the province shows a point-like aggregation growth. From the land use transfer matrix, the expansion of construction land is shown to crowd out the space occupied by arable land, which is consistent with the results of the study by Zhou et al. [35]. Land is exploited for economic benefits, and the structure of land use changes as a result of environmental influences directly affects ecosystem values [36,37].

Land development and utilization should adhere to the following principles: (1) All economic activities requiring land approval must align with the guidelines of the "three zones and three lines", utilizing land judiciously and efficiently within the urban development boundary. The timely replenishment of land occupied by urban development and industrial construction is imperative. All relevant departments are required to enhance monitoring and early warning mechanisms, establish a comprehensive system for the lifelong investigation of responsibilities pertain to ecological environment damage [38], and allow ecological environmental protection to be linked to political performance, so as to promote the intensive and efficient use of land for construction. (2) Water, as the land category providing the highest ESV, should be better regulated. Particular attention should be paid to the protection of the province's coastline, and Dongying should strengthen the

ecological protection and restoration of the Yellow River Delta. (3) Mountainous areas such as the Luzhong mountainous area should adhere to the policies of returning farmland to forests, improve the ecological environment, strengthen supervision, organize relevant departments to "look back" on a regular basis, consolidate the results of the work, and avoid replanting in areas where farmland has been returned to forests. (4) Shandong Province is the main grain-producing area in China, and the phenomenon of the crowding out of arable land by construction land has been obvious in the past two decades. The ESV provided by arable land in 2020 is CNY 2.266 billion less than that in 2000.

The protection of arable land and permanent basic farmland should be strengthened with the help of policy, legal, scientific, and technological measures to replenish arable land resources, deal with the problem of land abandonment, carry out farmland water conservancy upgrading projects, popularize good seeds, and increase food production to guarantee food security and income.

### 4.2. ESV Estimates and Spatial and Temporal Distribution Patterns

The estimation of ESV in this paper adopts the equivalent factor method, which has the advantages of requiring less data and being intuitive and easy to operate compared with the functional price method, and it is suitable for the assessment of the ecosystem service value at different scales [30,39]. The results of this study are similar to those of Wang et al. [40] and Guan et al. [41] in the same study area. Although the ESV correction coefficients vary, they do not affect the quantitative relationship between the spatially analyzed and studied ESV values for each period. Considering the complexity, variability, dynamics, and other characteristics of ecosystems, these give the equivalent factor method approach some limitations [42]. Subsequent research can be continued in the following areas: First, the area and equivalent factor of each type of land use directly affects the accuracy of the ESV measurement, and higher-resolution land use remote sensing images are needed. Second, in terms of analyzing the impact of LUCC on ESV from the spatial perspective, this paper does not go far enough. Subsequent methods such as land use change mapping, ESV change hot spot combination, and global and local spatial autocorrelation have been used to demonstrate the relationship between ESV and LUCC from a spatial perspective [5,19]. In view of these shortcomings, subsequent in-depth research should be carried out. Technical support is given to provide effective suggestions for land optimization in different counties of Shandong Province.

### 5. Conclusions

This paper explores the impact of land use change on ESV and the dynamic evolution of its spatial pattern in Shandong Province at a grid scale of 10 km × 10 km. The main conclusions are as follows: (1) From 2000 to 2020, land use change is characterized by an increase in construction areas and water, and a decrease in unused land, forestland, grassland, and cultivated land areas. Cultivated land was converted into construction land, and grassland and construction land partially transformed into cultivated land in Shandong Province. The changes in land use types were more pronounced in the first decade compared to the second decade. (2) During the study period, the highest ESV among the primary service functions was observed in the regulating services, with hydrological regulation providing the highest ESV among the eleven secondary service categories. The areas of high ESV are concentrated in the Southern Four Lakes in Shandong Province and around Laizhou Bay. There is a low ESV around the mountainous areas in central Shandong Province. (3) Within the three time periods, cultivated land, grassland, and unused land provided a higher negative contribution rate, while water provided a higher positive contribution rate.

**Author Contributions:** Conceptualization, Y.L. and T.L.; methodology, T.L. and D.S.; software, S.J.; validation, T.L., D.S. and S.J.; formal analysis, T.L.; investigation, H.Y.; resources, T.L.; data curation, S.J.; writing—original draft preparation, T.L.; writing—review and editing, T.L.; visualization, Y.L.;

supervision, H.Y.; project administration, Y.L.; funding acquisition, H.Y. All authors have read and agreed to the published version of the manuscript.

**Funding:** This research was funded by the project National Science and Technology Basic Resources Survey Special Project: Investigation on Urbanization and Infrastructure along the China-Mongolia-Russia Economic Corridor (2017FY101303), Shandong Provincial Natural Science Foundation, China (ZR2021MG040), and Shandong Provincial Humanities and Social Project, China (2021-JCGL-01).

**Institutional Review Board Statement:** Not applicable.

**Informed Consent Statement:** Not applicable.

**Data Availability Statement:** The data presented in this study are available on request from the corresponding author.

**Conflicts of Interest:** The authors declare no conflicts of interest.

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
