# Peer review of "Analysis of Spatial—Temporal Variation in Ecosystem Service Value in Shandong Province over the Last Two Decades"

_sustainability, doi:10.3390/su16020515_

Round 1
Reviewer 1 Report
Comments and Suggestions for Authors
The study is about the value of ecosystem services in a Province in China. The study examines the changes in the values of ecosystem services and highlights the hotspots and cold spots. It is an interesting study. The authors can consider the following in improving the manuscript.
1. The image classification method adopted by the study is not state in the text. It is important to state the classification method and the accuracies of the classifications of each year.
2. The equations cited in equations 1 and 2 are not found in the reference "29". Are the equations, especially the attitudinal model, developed by the authors. The authors should ensure that the equations and models are properly referenced unless they are developed by the authors.
3. In the same vein, the authors have not explained the terms of equation 7.
4. The discussion section is mainly about the limitations of the study. It can be improved by explaining the results in the context of the existing literature.
5. The manuscript needs moderate copy-editing. For example, "The scales are often dependent" is like a truncated sentence without stating what the scales are dependent on. Also, "has lacked of spatial representation" should be changed.
Comments on the Quality of English LanguageModerate editing of English language required
Author Response
Responses to Reviewer 1
Question 1.
The image classification method adopted by the study is not state in the text. It is important to state the classification method and the accuracies of the classifications of each year.
Answers: Thank you for your suggestion. The remote sensing images for the manuscript are from the Resource and Environmental Science Data Center of the Chinese Academy of Sciences (http://www.resdc.c-n), download the data for 2000, 2010 and 2020.
Question 2.
The equations cited in equations 1 and 2 are not found in the reference "29". Are the equations, especially the attitudinal model, developed by the authors. The authors should ensure that the equations and models are properly referenced unless they are developed by the authors.
Answers: We appreciate your suggestions and sincerely apologize for the occurrence of this error. Please refer to references 5 and 30 for clarification. In view of this problem, we have reexamined the full text and made corrections. We will resolutely avoid such problems.
[5] Yang Guangzong, Lyu Kai, Li Feng. Spatial and Temporal Correlation Analysis of Land Use Change and Ecosystem Service Value in Nanchang City Based on Grid Scale[J]. China Land Science. 2022, 36(08): 122-130.
[30] Han Huiran, Yang Chengfeng, Song Jinping. The Spatial-Temporal Characteristic of Land Use Change in Beijing and Its Driving Mechanism[J]. Economic Geography, 2015, 35(05): 148-154, 197.
Question 3.
In the same vein, the authors have not explained the terms of equation 7.
Answers: To address this issue, we looked for related literature and found that hotspot analysis is used in the following two similar papers, indicating that this method can be used to study such problems. Also, references have been added and explanations of formulas have been added to the manuscript.
Guo Chunyang, Gao Shang, Zhou Boyan, et al. Effects of land use change on ecosystem service value in Funiu Mountain based upon a grid square[J]. Acta Ecologica Sinica, 2019, 39(10): 3482-3493.
Zhang Fa, Yusufujiang Rusuli,Aierken Tuersun. Spatio-temporal change of ecosystem service value in Bosten Lake Watershed based on land use[J]. Acta Ecologica Sinica, 2021, 41(13): 5254-5265.
Question 4.
The discussion section is mainly about the limitations of the study. It can be improved by explaining the results in the context of the existing literature.
Answers: Thank you for your suggestion, and we agree with you. Through extensive reading of the literature, we have made changes to the discussion section, which is highlighted in the text.
Question 5.
The manuscript needs moderate copy-editing. For example, "The scales are often dependent" is like a truncated sentence without stating what the scales are dependent on. Also, "has lacked of spatial representation" should be changed.
Answers: We have corrected these mistakes. Firstly, Scales tend to be scale-dependent, and conducting ESV assessments at these scales has the disadvantage of under-representing spatial information. The second, after reviewing the literature, the original manuscript was found to be inaccurately expressed and was therefore deleted.

Reviewer 2 Report
Comments and Suggestions for Authors
This is well written paper that adds a useful case to the literature and deserves publication. I noted a typo in Fig. 4 (uesd instead of used). Also, by construction land, do you mean land devoted to urban uses ? IT would be useful to clarify what is included as the term is not commonly used in the US.
It would be helpful to add a map early in the paper showing the location of Shandong Province in the PRC and also showing the location of Shandong's major cities. This would help orient readers who may not be entirely familiar with Shandong Province.
Comments on the Quality of English LanguageSee above- two minor issues.
Author Response
Responses to Reviewer 2
Question 1.
This is well written paper that adds a useful case to the literature and deserves publication. I noted a typo in Fig. 4 (uesd instead of used). Also, by construction land, do you mean land devoted to urban uses ? IT would be useful to clarify what is included as the term is not commonly used in the US.
Answers: Thank you for your valuable suggestion. It was indeed an oversight on our part. We have redrawn the figures, as illustrated below, and made the necessary modifications in the manuscript. In response to the question of the meaning of construction land, we offer the following explanation:The article adopts the China Multi-Period Land Use Remote Sensing Monitoring Dataset (CNLUCC), which adopts a two-tier classification system: the first tier is divided into six categories, which are divided into cultivated land, forestland, grassland, water, construction land, and unused land, mainly based on the attributes of the land resources and their utilization; and the second tier is divided into 23 types, mainly based on the natural attributes of the land resources. Among them, construction land refers to urban and rural settlements and the industrial, mining, transportation and other land outside of them, including urban land, rural settlements and other construction land.
Original Figure 4
Question 2.
It would be helpful to add a map early in the paper showing the location of Shandong Province in the PRC and also showing the location of Shandong's major cities. This would help orient readers who may not be entirely familiar with Shandong Province.
Answers: Thank you for your suggestion, we have added an image of the research area as shown below.

Reviewer 3 Report
Comments and Suggestions for Authors
This research focused on the analysis of spatial-temporal variation of ecosystem services value in Shandong Province over the past two decades, which will provide scientifical reference for Shandong Province to promote the sustainable development of the ecosystem and economy.
Comments and Suggestions for Authors are as follows.
1. Keywords section, more keywords should be selected.
2.Introduction section, Literature review needs improvement. The shortcomings of the existing research are suggested to be highlighted and the significance of this study should be further clarified. Additionally, It is also important to add some relevant studies about ecosystem services in Shandong Province.
3. Data sources section, the information (eg, spatial resolution) of remote sensing image data should be added in this part.
4. P2-P4: in the sentence “……, 1762 grids 10km10km were created based on three periods of ……”, “10km10km” should be changed into “10km×10km”. Moreover, there is some mistake in the sentence “where is the economic value of the function of providing food production services per unit area of farmland ecosystem (CNY/hm); is the type of farmland crop; is the area planted with the seed crop (hm); is the national average price of the seed crop in a given year (CNY/t); is the yield per unit area of the seed crop (t/hm); and is the area planted with all crops (hm)”. “hm” should be changed into “hm2”.
5. In this study, please explain the basis for selecting "10km×10km" as the basic analysis unit.
6. Table 2 and Table 3 need to be further improved.
7. Disscusion and Conclusion section, I suggest that the discussion be separated from conclusion section. There is not enough literature reviews and research summaries in the section, which should be strengthened appropriately.
8. Conclusions section, it should be more concise and condensed. I suggest that if the author summarize the laws and enlightenment behind it based on the research results.
9.The languages of this manuscript need to be further improved.

Comments on the Quality of English LanguageThis research focused on the analysis of spatial-temporal variation of ecosystem services value in Shandong Province over the past two decades, which will provide scientifical reference for Shandong Province to promote the sustainable development of the ecosystem and economy.
Comments and Suggestions for Authors are as follows.
1. Keywords section, more keywords should be selected.
2.Introduction section, Literature review needs improvement. The shortcomings of the existing research are suggested to be highlighted and the significance of this study should be further clarified. Additionally, It is also important to add some relevant studies about ecosystem services in Shandong Province.
3. Data sources section, the information (eg, spatial resolution) of remote sensing image data should be added in this part.
4. P2-P4: in the sentence “……, 1762 grids 10km10km were created based on three periods of ……”, “10km10km” should be changed into “10km×10km”. Moreover, there is some mistake in the sentence “where is the economic value of the function of providing food production services per unit area of farmland ecosystem (CNY/hm); is the type of farmland crop; is the area planted with the seed crop (hm); is the national average price of the seed crop in a given year (CNY/t); is the yield per unit area of the seed crop (t/hm); and is the area planted with all crops (hm)”. “hm” should be changed into “hm2”.
5. In this study, please explain the basis for selecting "10km×10km" as the basic analysis unit.
6. Table 2 and Table 3 need to be further improved.
7. Disscusion and Conclusion section, I suggest that the discussion be separated from conclusion section. There is not enough literature reviews and research summaries in the section, which should be strengthened appropriately.
8. Conclusions section, it should be more concise and condensed. I suggest that if the author summarize the laws and enlightenment behind it based on the research results.
9.The languages of this manuscript need to be further improved.
Author Response
Responses to Reviewer 3
Question 1.
Keywords section, more keywords should be selected.
Answers: We have added two keywords "spatial and temporal distribution" and "spatial variation" to the manuscript.
Question 2.
Introduction section, Literature review needs improvement. The shortcomings of the existing research are suggested to be highlighted and the significance of this study should be further clarified. Additionally, It is also important to add some relevant studies about ecosystem services in Shandong Province.
Answers: Thank you for your professional advice. First of all, we have added the shortcomings of the existing literature in the literature review section, which has been highlighted in the article. Secondly, the last paragraph of the introduction section has a statement of the significance of the study: In order to provide scientific reference for the rational utilization of land, protection and restoration of ecosystems, and to provide scientific support for the promotion of ecological and economic sustainable development and the enhancement of people's well-being in Shandong Province.
Question 3.
Data sources section, the information (eg, spatial resolution) of remote sensing image data should be added in this part.
Answers: We have added relevant specifics to the manuscript as follows: Spatial resolution of 1 km × 1 km.
Question 4.
P2-P4: in the sentence “……, 1762 grids 10km10km were created based on three periods of ……”, “10km10km” should be changed into “10km×10km”. Moreover, there is some mistake in the sentence “where is the economic value of the function of providing food production services per unit area of farmland ecosystem (CNY/hm); is the type of farmland crop; is the area planted with the seed crop (hm); is the national average price of the seed crop in a given year (CNY/t); is the yield per unit area of the seed crop (t/hm); and is the area planted with all crops (hm)”. “hm” should be changed into “hm2”.
Answers: Thank you for your valuable suggestion, we have changed "10km10km" to "10km × 10km". And we have also changed "hm" to "hm2".
Question 5.
In this study, please explain the basis for selecting "10km×10km" as the basic analysis unit.
Answers: Referring to the literature of Yang Guangzong and other literature, with the help of ArcGIS 10.2 software Greate Fishnet, Dissolve, Clip, Merge and other analyzing tools, and combining with the study area, we finally decided to adopt "10km×10km" as the basic research unit.
Question 6.
Table 2 and Table 3 need to be further improved.
Answers: Tables 2 and 3 are modified as follows:
Table 2. Dynamic changes of land use in Shandong Province from 2000 to 2020.
Year |
Cultivated land |
Forestland |
Grassland |
Water |
Construction land |
Unused land |
||||||
area /hm2 |
Proportions/% |
area /hm2 |
proportions/% |
area /hm2 |
proportions/% |
area /hm2 |
proportions/% |
area /hm2 |
proportions/% |
area /hm2 |
proportions/% |
|
2000-2010 |
-106078 |
-0.70 |
-58622 |
-0.38 |
-533882 |
-3.44 |
94912 |
0.61 |
743247 |
4.79 |
-136436 |
-0.88 |
2010-2020 |
-176216 |
-1.14 |
3838 |
0.03 |
-3489 |
-0.02 |
-1745 |
-0.02 |
188429 |
1.21 |
-8724 |
-0.06 |
2000-2020 |
-282294 |
-1.84 |
-54784 |
-0.35 |
-537371 |
-3.46 |
93168 |
0.59 |
931676 |
6.00 |
-145160 |
-0.94 |
Table 3. Values and changes of ecosystem service in Shandong Province from 2000 to 2020. (108CNY、%)
Ecosystem Services |
2000 |
2010 |
2020 |
||||||
ESV |
Percentage
|
ESV
|
Percentage
|
Change 2000-2010 ESV |
ESV
|
Percentage
|
Change 2000-2010 ESV |
||
Provisioning Services |
FS |
197.17 |
6.69 |
193.82 |
6.56 |
-3.35 |
190.80 |
6.50 |
-3.02 |
RMS |
108.54 |
3.68 |
10.03 |
3.52 |
-4.50 |
102.63 |
3.49 |
-1.41 |
|
WS |
101.41 |
3.44 |
109.32 |
3.7 |
7.90 |
109.06 |
3.71 |
-0.25 |
|
Regulating Services |
GR |
227.90 |
7.73 |
213.84 |
7.24 |
-14.06 |
211.47 |
7.20 |
-2.37 |
CR |
313.27 |
10.6 |
277.58 |
9.4 |
-35.68 |
276.39 |
9.41 |
-1.19 |
|
EP |
154.45 |
5.24 |
149.23 |
5.05 |
-5.22 |
148.73 |
5.06 |
-0.50 |
|
HR |
1205.35 |
40.9 |
1294.85 |
43.8 |
89.50 |
1291.81 |
43.98 |
-3.03 |
|
Supporting Services |
SC |
321.77 |
10.9 |
304.19 |
10.3 |
-17.58 |
300.55 |
10.23 |
-3.64 |
MN |
33.13 |
1.12 |
31.69 |
1.07 |
-1.44 |
31.27 |
1.06 |
-0.42 |
|
CB |
185.47 |
6.29 |
178.28 |
6.04 |
-7.19 |
177.69 |
6.05 |
-0.59 |
|
Cultural services |
AL |
98.42 |
3.34 |
97.16 |
3.29 |
-1.25 |
96.86 |
3.30 |
-0.31 |
Total |
2946.86 |
100 |
2953.99 |
100 |
14.26 |
2937.26 |
100.00 |
-33.46 |
Question 7.
Disscusion and Conclusion section, I suggest that the discussion be separated from conclusion section. There is not enough literature reviews and research summaries in the section, which should be strengthened appropriately.
Answers: Thanks to your suggestions, we have read a great deal of the literature and revised the manuscript to divide the discussion and conclusions into two chapters. The following literature has been added to enrich and complete the discussion section.
Zhou Y, Li X, Liu Y. Land use change and driving factors in rural China during the period 1995-2015[J]. Land Use Policy, 2020, 99: 105048.
Zhang X, Ren W, Peng H. Urban land use change simulation and spatial responses of ecosystem service value under multiple scenarios: A case study of Wuhan, China[J]. Ecological Indicators, 2022, 144: 109526.
Fang Z, Ding T, Chen J, et al. Impacts of land use/land cover changes on ecosystem services in ecologically fragile regions[J]. Science of the Total Environment, 2022, 831: 154967.
WANG Dan, JING Yande, HAN Shanmei. Research on the spatial coupling relationship between land-use patternand ecosystem services valuein Nansi Lake basin based upon a grid square[J]. Journal of Qufu Normal University. 2023, 49(1):96-105.
GUAN Mei ,ZHANG Wenxin ,JIANG Haiming, et al. Cross-sensitivity evaluation of ecosystem services based onland use changes in Shandong Province[J]. Journal of China Agricultural University. 2022,27(6):192-203
Question 8.
Conclusions section, it should be more concise and condensed. I suggest that if the author summarize the laws and enlightenment behind it based on the research results.
Answers: Thanks to your professional advice, we have revised the conclusion section of the manuscript. The conclusions are more concise than before.
Question 9.
The languages of this manuscript need to be further improved.
Answers: We tried our best to improve the manuscript and made some changes to the manuscript. These changes will not influence the content and framework of the paper. And here we did not list the changes but marked in yellow in the revised paper.
